Coronatine inhibits stomatal closure and delays hypersensitive response cell death induced by nonhost bacterial pathogens

Lee Seonghee 1
Ishiga Yasuhiro 1
Clermont Kristen 1 2
Mysore Kirankumar S. ksmysore@noble.org 1
1 The Samuel Roberts Noble Foundation , Plant Biology , Ardmore, Oklahoma , USA
Lubberstedt Thomas
2 Current address: Virginia Tech, Department of Biological Sciences, Blacksburg, VA, USA.

Electronic publication date: 2013 Feb 12
Publication date: 2013
Volume: 1
Electronic Location ID: e34
Received 2012 Nov 23; Accepted 2013 Jan 16
Copyright: © 2013 Lee et al.
Copyright year: 2013
Copyright holder: Lee et al.
License: This is an open access article distributed under the terms of the Creative Commons Attribution License, which permits unrestricted use, distribution, and reproduction in any medium, provided the original author and source are credited.
License URL: https://creativecommons.org/licenses/by/3.0/

Keywords: Coronatine, Nonhost resistance, Hypersensitive response, Nicotiana benthamiana

Funding: The Samuel Roberts Noble Foundation This work was funded by the Samuel Roberts Noble Foundation. The funders had no role in study design, data collection and analysis, decision to publish, or preparation of the manuscript.

==============================
Pseudomonas syringae is the most widespread bacterial pathogen in plants. Several strains of P. syringae produce a phytotoxin, coronatine (COR), which acts as a jasmonic acid mimic and inhibits plant defense responses and contributes to disease symptom development. In this study, we found that COR inhibits early defense responses during nonhost disease resistance. Stomatal closure induced by a nonhost pathogen, P. syringae pv. tabaci, was disrupted by COR in tomato epidermal peels. In addition, nonhost HR cell death triggered by P. syringae pv. tabaci on tomato was remarkably delayed when COR was supplemented along with P. syringae pv. tabaci inoculation. Using isochorismate synthase (ICS)-silenced tomato plants and transcript profiles of genes in SA- and JA-related defense pathways, we show that COR suppresses SA-mediated defense during nonhost resistance.

Introduction

Plants possess a natural innate immune system that efficiently detects potential pathogens. The initial stage of this defense system is based on the perception of pathogen- or microbe-associated molecular patterns (PAMPs or MAMPs) through pattern recognition receptors present at the plant cell surface (Boller & Felix, 2009). Recent studies suggest that plant stomata can play an active role as part of the plant innate immune system in restricting bacterial invasion (Melotto, Underwood & He, 2008; Melotto et al., 2006). Perception of multiple bacterial PAMPs, including flagellin, lipopolysaccharide (LPS) and elongation factor Tu (EF-Tu), induced closure of stomata in epidermal peels of Arabidopsis leaves (Melotto et al., 2006). Moreover, a significant induction of stomatal closure was observed within the first hour of contact with both host and nonhost bacterial pathogens (Melotto, Underwood & He, 2008). However, bacterial pathogens have evolved to acquire specific virulence factors such as coronatine (COR) to overcome PAMP-triggered immunity (PTI) and stomata-based defense (Underwood, Melotto & He, 2007). For example, the virulent pathogen P. syringae pv. tomato DC3000 produces COR three hours after infection in the apoplast and on the plant surface to reopen closed stomata, thus allowing more bacteria to enter (Melotto, Underwood & He, 2008; Melotto et al., 2006; Underwood, Melotto & He, 2007). In addition, COR enhances bacterial multiplication in the apoplast and the formation of necrotic lesions (cell death) surrounded by chlorotic halos, thus promoting systemic susceptibility (Bender, 1999; Cui et al., 2005; Preston, Bertrand & Rainey, 2001). Chlorosis associated with disease caused by several pathovars of P. syringae has been attributed mainly to the phytotoxin COR (Bender et al., 1987; Zhao et al., 2003).

COR consists of the polyketide coronafacic acid (CFA) (Brooks, Bender & Kunkel, 2005) and coronamic acid (CMA), a cyclized derivative of isoleucine (Mitchell, 1985). COR has structural and functional similarity to jasmonates and jasmonic acid-isoleucine (JA-Ile) (Katsir et al., 2008; Uppalapati et al., 2005; Weiler et al., 1994). COR contributes to the virulence of P. syringae pv. tomato DC3000 in Arabidopsis, tomato, collards (Brassica oleracea) and turnip (Brooks et al., 2004; Zhao et al., 2003). By mimicking jasmonates, COR stimulates the JA pathway in Arabidopsis and tomato, and thereby functions to suppress the SA pathway and/or closure of stomata, thus allowing bacteria to reach higher densities in planta (Melotto et al., 2006; Uppalapati et al., 2005; Zhao et al., 2003). In a recent study, Zheng et al. (2012) reported that COR disrupts the accumulation of the important plant defense hormone salicylic acid (SA) for stomatal reopening and bacterial propagation in both local and systemic tissues of Arabidopsis. However, it is not clear whether COR is also involved in promoting entry of nonhost bacterial pathogens via stomata and nonhost bacterial growth at the initial stage of infection.

P. syringae pv. tabaci is the causal agent for the wild fire disease in tobacco (Uchytil & Durbin, 1980). This pathogen does not produce COR and is unable to infect nonhost plant species such as tomato. It has been noted that most of the stomata in epidermal peels of tomato remained closed after three hours incubation with nonhost pathogen P. syringae pv. tabaci, while virulent pathogen P. syringae pv. tomato DC3000 producing COR is able to reopen stomata after three hours treatment in tomato epidermal peels (Melotto et al., 2006). In the current study, we examined how COR can affect the early defense responses including stomatal defense and nonhost HR cell death by a nonhost pathogen in tomato. In addition, we determined the expression of defense genes related to SA and JA pathways after treatment of host or nonhost pathogens along with COR. Our findings from this study clearly show that COR confers the virulence activity to suppress early plant defense systems against nonhost bacterial pathogens in tomato.

Materials and Methods

Plant and pathogen materials

Tomato (Solanum lycopersicum cv. Glamour) plants were grown on 10 cm diameter round pots containing potting soil (BM7) (Berger Co., Quebec, Canada) and maintained in the greenhouse under a 14-h light/10-h dark photoperiod at 23 ± 2 °C. Four-week-old tomato plants were used for the experiments. P. syringae pv. tomato DC3000 and P. syringae pv. tomato T1 were used as virulence bacterial pathogens (host pathogens), and P. syringae pv. tabaci and P. syringae pv. phaseolicola were used as nonhost bacterial pathogens on tomato.

These host and nonhost bacterial pathogens were grown overnight at 28 °C in King’s B (KB) medium [10 g Proteose pepton #2 (BD Difco, New Jersey, USA); 1.5 g anhydrous K2HPO; 15 g glycerol; 5 mL MgSO4 (1M sterile)] containing appropriate antibiotics (kanamycin 25 mg/ml and rifamycin 50 mg/ml). After the overnight culture, bacterial cells were centrifuged at 5,000 rpm for 10 min, and the cell pellet was resuspended in 10 ml of sterile distilled water. The concentration of bacteria in the culture suspension was measured in a spectrophotometer (the optical density at OD 600 nm) and diluted to appropriate concentrations for inoculations.

Bacterial pathogen inoculation

Bacterial suspensions (3 × 107 colony forming unit [CFU]/ml) were prepared in sterile distilled water containing 0.025% Silwet L-77 (OSI Specialties Inc., Danbury, CT, USA) and sprayed on tomato plants. The inoculated plants were then incubated in growth chambers at 90 to 100% RH for the first 24 h. The inoculated plants were observed at 7 days post inoculation (dpi) for symptom development. Bacterial growth in leaves was measured by determining the internal bacterial population. Prior to sampling, leaves were surface-sterilized with 15% H2O2 for 3 min to eliminate epiphytic bacteria and then washed with sterile distilled water. The leaves were then homogenized in sterile distilled water, and serial dilutions were plated onto KB medium containing antibiotics (kanamycin, 25 mg/ml). The bacterial population at 0 dpi was estimated from leaves harvested 1 h post inoculation (hpi). Bacterial growth patterns at 0, 1 and 3 dpi were evaluated in two independent experiments (four leaf samples per each experiment).

Stomata assay

The bacterial culture of P. syringae pv. tabaci was centrifuged and resuspended in distilled water at a concentration of 5 × 104 CFU/ml. For stomatal closure assay, the tomato epidermal peel was prepared as described (Melotto et al., 2006). Epidermis of tomato leaves were peeled off and immediately floated on stomata opening buffer (10 mM MES-KOH, 30 mM KCl, pH 6.3). After confirming stomata opening under the microscope, the epidermal peels were treated with P. syringae pv. tabaci with or without COR (100 ng/ml; obtained from C. Bender, Oklahoma State University). Approximately 100 random stomatal apertures were measured per each treatment, and three samples were collected from each experiment.

Inhibition of nonhost HR cell death by COR and bacterial growth determination

To determine if COR inhibits nonhost HR cell death, two nonhost bacterial pathogens, P. syringae pv. tabaci and P. syringae pv. phaseolicola were used for this experiment in tomato. Bacterial cultures grown in KB medium overnight at 28 °C were centrifuged (3,500 rpm, 10 min) and resuspended in MES buffer (MES 10 mM, pH 6.5). The bacterial suspension was infiltrated into fully expanded tomato leaf using a 1.5 ml needleless syringe for examining bacterial growth and nonhost HR cell death. Three different concentrations (100, 200 and 300 ng/ml) of COR were tested to determine COR toxicity in tomatao leaves. Both 200 and 300 ng/ml showed visible cell death and chlorosis symptoms. The concentration of 100 ng/ml of COR was added to bacterial suspension (104 CFU/ml) to determine the inhibition of HR cell death. Infiltrated leaves were inspected for HR cell death after 24 h and 48 h inoculation. Tomato plants silenced for isochorismate synthase (ICS) accumulation by RNAi (ICS-RNAi) (Uppalapati et al., 2007) were also used to evaluate if COR inhibits SA accumulation and suppress HR cell death. To determine whether the bacterial growth is promoted by COR, a low concentration of P. syringae pv. tabaci (2 × 102 CFU/ml) was inoculated with or without COR. The bacterial population in the apoplast was examined at 0, 1 and 3 dpi.

The detection of H2O2 in leaf tissues was done by 3,3’-diaminobenzidine (DAB) staining. H2O2 reacts with DAB to form a reddish-brown stain. P. syringae pv. tomato DC3000 and P. syringae pv. tabaci were infiltrated into one side of the tomato leaf with or without COR (100 ng/ml; obtained from C. Bender, Oklahoma State University) (Li et al., 2005; Zheng et al., 2012)). The DAB staining procedure was followed as described (Rojas et al., 2012).

RNA isolation and quantitative real-time PCR

Total RNA was purified from tomato leaves infiltrated with water (mock control), COR, nonhost pathogen P. syringae pv. tabaci (Psta), Psta with COR, or host pathogen P. syringae pv. tomato T1 (Pst T1). Total RNA was extracted using TRIzol (Invitrogen), and two treated or inoculated leaves were pooled to represent one biological replicate. Total RNA was treated with DNase I (Invitrogen), and 1 µg RNA was used to generate cDNA using Superscript III reverse transcriptase (Invitrogen) and oligo d(T)15–20 primers. The cDNA (1 : 20) was then used for qRT-PCR using Power SYBR Green PCR master mix (Applied Biosystems, Foster City, CA, USA). Primers specific for tublin 4 was used to normalize small differences in template amounts. To determining COR-mediated inhibition of SA and JA hormonal pathway, real-time quantitative PCR was performed with primers shown in Supplemental Table 1. Average Cycle Threshold (CT) values calculated using Sequence Detection Systems (version 2.2.2; Applied Biosystems) from duplicate samples were used to determine the fold expression relative to controls. All qRT-PCR were performed using ABI PRISM 7700 Sequence Detection System (Applied Biosystems, CA, USA) and calculation was made according to the company manual (User Bulletin #2).

Results and Discussion

Pseudomonas syringae pv. tabaci inoculation on tomato induces rapid closure of stomata and nonhost HR

The bacterial pathogen P. syringae pv. tomato DC3000 causes bacterial speck disease on a host plant, tomato. Five days after spray inoculation of P. syringae pv. tomato DC3000, a number of bacterial leaf spots at infection sites were observed in tomato. In contrast, nonhost bacterial pathogen P. syringae pv. tabaci that is the causal agent for wildfire leaf disease in tobacco is unable to infect tomato (Supplemental Fig. S1A). At the initial infection process, bacterial pathogens produce virulence effectors to disrupt early plant defense systems such as the stomata-based immunity (Melotto et al., 2006; Melotto, Underwood & He, 2008) and the HR cell death at the site of infection (Underwood, Melotto & He, 2007). Stomata are the active barriers in limiting bacterial entry into apoplast. Several recent studies showed that stomatal closure induced by bacterial pathogens or bacterial PAMPs is a common plant defense mechanism in plants (Melotto et al., 2006; Cui et al., 2005; Bender, 1999). We also found that most stomata on tomato epidermal peels remained closed after inoculation of a nonhost pathogen, P. syringae pv. tabaci (Fig. 1).

Figure 1 COR suppresses stomatal closure induced by P. syringae pv. tabaci in tomato (A and B).

To determine suppression of stomatal closure by COR, tomato epidermal peels were floated on stomata opening buffer (KCl 30 mM; pH 6.3) and stomata were observed under a light microscope to assure stomatal opening. The epidermal peels were incubated with P. syringae pv. tabaci suspension (5 × 104 CFU/ml) with or without COR (100 ng/ml) in the bacterial suspension for three hours. Images (A) were taken using a light microscope, and individual stomatal apertures were measured (B) using ImageJ software (http://rsb.info.nih.gov/ij/). Approximately 50 stomata were counted for each epidermal peel, and a total of three peels were used for the analysis of each treatment. Bars represent the means ± standard deviation (SD) from two independent experiments. Different letters above the bars indicate significantly different values between treatments (P < 0.05, Student’s t test).

The nonhost HR cell death at the site of infection is the most common defense mechanism observed in many plants in response to nonhost bacterial pathogens (Mysore & Ryu, 2004). Supplemental Fig. S1 shows that when nonhost pathogen P. syringae pv. tabaci was infiltrated on tomato leaf, nonhost HR cell death was observed at the site of inoculation, while host pathogen P. syringae pv. tomato DC3000 suppressed HR cell death (Fig. S1B).

COR produced from P. syringae pv. tomato DC3000 plays an important role to disrupt early plant defense responses including stomatal closure (Melotto et al., 2006; Melotto, Underwood & He, 2008). We speculated that COR can suppress stomata closure induced by a nonhost pathogen and nonhost HR cell death-induced P. syringae pv. tabaci in tomato. In this study, we performed experiments to examine whether COR plays a role in inhibiting the early defense responses against nonhost pathogen P. syringae pv. tabaci in tomato.

Coronatine inhibits stomatal closure induced by nonhost pathogen, P. syringae pv. tabaci

Phytopathogens have evolved specific virulence factors to overcome early plant defense responses. In the case of virulent pathogen P. syringae pv. tomato strain DC3000, three hours after infection, bacteria produce the diffusible virulence factor coronatine (COR) in the apoplast and on the plant surface to reopen closed stomata, allowing increased bacterial entry (Melotto, Underwood & He, 2008; Melotto et al., 2006; Underwood, Melotto & He, 2007). However, most stomata remained closed in tomato when inoculated with a nonhost pathogen, P. syringae pv. tabaci (Fig. 1). To determine if COR can suppress the stomatal closure induced by a nonhost pathogen, we added COR (100 ng/ml) to P. syringae pv. tabaci suspension, and tomato epidermis was incubated with this suspension for three hours. Interestingly, most stomata remained open, and the aperture size of open stomata was similar to that of an aperture of stomata incubated with stomata opening buffer (Fig. 1). These findings indicate that COR acts as a virulence factor to disrupt stomatal closure induced by nonhost pathogen P. syringae pv. tabaci in tomato.

Coronatine delays nonhost HR cell death induced by nonhost pathogen P. syringae pv. tabaci

In addition to the COR-mediated inhibition of stomatal closure, we also examined whether COR can suppress nonhost HR cell death induced by P. syringae pv. tabaci. It has been known that COR is not a host-specific phytotoxin (Uppalapati et al., 2005), but it has never been reported whether COR can delay or suppress HR cell death in plants. As shown in Supplemental Fig. S1, the inoculation of P. syringae pv. tabaci induced typical nonhost HR cell death on tomato leaves within 24 h after infiltration (hpi), while host pathogen P. syringae pv. tomato DC3000 did not induce any visible cell death at that time point. We speculate that the nonhost HR in tomato is triggered either by PAMPs or effectors of nonhost pathogen P. syringae pv. tabaci. Such PAMP- and/or effector-triggered HR is probably suppressed during P. syringae pv. tomato DC3000-tomato interaction.

Interestingly, nonhost HR cell death was not observed at 24 hpi when P. syringae pv. tabaci was infiltrated along with COR in contrast to infiltration of P. syringae pv. tabaci alone (Fig. 2A). At 48 hpi, even though the nonhost HR cell death was visible at both infiltration sites, the cell death caused by infiltration of P. syringae pv. t a b a c i + COR was milder compared to P. syringae pv. tabaci alone. This result clearly indicates that COR delays nonhost HR cell death caused by a nonhost bacterial pathogen in tomato. However, the COR-mediated cell death suppression is not strong enough to completely suppress the HR caused by a nonhost pathogen. HR is associated with defenses that are manifested by development of rapid cell death through the SA-mediated pathway (Alvarez, 2000; Mysore & Ryu, 2004). It has been reported that COR inhibit SA accumulation in Arabidopsis (Zheng et al., 2012). To investigate the role of SA in delaying nonhost HR cell death, we examined nonhost HR cell death in ICS-silenced tomato plants (ICS-RNAi); (Uppalapati et al., 2007). ICS is required for SA biosynthesis in plants. The nonhost HR cell death was not detected 24 h after infiltration of P. syringae pv. tabaci alone or P. syringae pv. tabaci + COR. Mild HR was observed 48 h after infiltration with P. syringae pv. tabaci alone or P. syringae pv. tabaci + COR in ICS-RNAi and was much weaker than the HR observed in non-silenced wild-type tomato plants. Interestingly, no difference was found between infiltration of P. syringae pv. tabaci alone or with P. syringae pv. tabaci + COR (Fig. 2A). This result suggests that the delay of nonhost HR by COR is likely due to suppression of the SA pathway.

Figure 2 COR delays nonhost HR cell death induced by P. syringae pv. tabaci and P. syringae pv. phaseolicola in tomato.

(A) Wild-type and ICS-RNAi tomato leaves were infiltrated with P. syringae pv. tabaci (6 × 102 CFU/ml) with or without COR (100 ng/ml). The development of nonhost HR cell death was determined 24 h and 48 h after infiltration. (B) Another nonhost pathogen, P. syringae pv. phaseolicola, (6 × 102 CFU/ml) was syringe (needleless) infiltrated with or without COR (100 ng/ml) to tomato leaf. After 24 h of infiltration, a photograph was taken (lower panel) and leaf samples were collected for DAB staining to visualize H2O2 production at the site of infection (upper panel). (C) Delay of nonhost HR cell death by COR producing nonhost pathogen P. syringae pv. tomato DC3000 in N. benthamiana. P. syringae pv. tomato DB29 is the mutant strain that does not produce COR. A photograph was taken 24 h after infiltration. (D) The bacterial population of P. syringae pv. tomato DC3000 (Pst DC3000), P. syringae pv. tabaci (Pstab) and P. syringae pv. tabaci with COR (Pstab + COR) was measured 0, 1 and 3 days post inoculation (dpi). The bacterial concentration of 2 × 102 CFU/ml was used for both pathogens. Bars represent the means ± standard deviation (SD) from three leaf samples per treatment. Three independent experiments were performed. Different letters above bars indicate significantly different values between treatments (P < 0.05, Student’s t test).

COR-mediated delay of nonhost HR cell death was also observed for another nonhost pathogen, P. syringae pv. phaseolicola (Fig. 2B), suggesting that the COR response was not specific to a particular pathogen. In addition, we also show that H2O2 accumulation (by DAB staining), a typical response during HR, was suppressed by COR (Fig. 2B). To further examine the delay of nonhost HR cell death by COR, we infiltrated N. benthamiana (a nonhost plant) with P. syringae pv. tomato DC3000 (103 CFU/ml) or P. syringae pv. tomato DB29 (103 CFU/ml), a mutant strain of DC3000 that does not produce COR (Uppalapati et al., 2007). After 24 h of infiltration, the level of nonhost HR cell death by P. syringae pv. tomato DB29 was much higher than that of P. syringae pv. tomato DC3000 (Fig. 2C). This finding clearly indicates that COR-producing P. syringae pathovars can delay nonhost HR cell death.

COR has been shown to promote compatible bacterial pathogen growth and develop disease symptoms in Arabidopsis and tomato (Melotto et al., 2006; Uppalapati et al., 2007; Zheng et al., 2012). Here we show that in tomato when COR was added to P. syringae pv. tabaci suspension prior to tomato leaf infiltration, it promoted the growth of nonhost pathogen (Fig. 2D). The level of bacterial growth of nonhost pathogen P. syringae pv. tabaci (with COR) was similar to the host pathogen P. syringae pv. tomato DC3000 at 1 dpi, but the bacterial population of P. syringae pv. tabaci was reduced at 3 dpi, suggesting that COR is not sufficient by itself to completely suppress the defense mechanism(s) of nonhost disease resistance. Alternatively, it is also possible that COR may not be stable in planta after 24 h and therefore was not able to further promote the growth of nonhost bacteria beyond 1 dpi.

P. syringae pv. tabaci does not produce COR and can’t infect tomato plants, but this pathogen is still functionally active to suppress defense responses in the host plant, tobacco. This indicates that P. syringae pv. tabaci produces unknown virulence factors to suppress PAMP-triggered immunity (PTI) in tobacco plants that may be different from the molecular mechanism of COR-dependent virulence. As shown in Fig. 2, COR can suppress nonhost HR cell death during initial phases of nonhost pathogen infection, but cell death eventually appeared after 24 hpi (Figs. 2A and 2C). This indicates other factors involved in defense mechanism of nonhost resistance are functionally regulated. It has been known that reactive oxygen species (ROS) and nitric oxide (NO) are important for HR cell death induced upon incompatible pathogen infection in plants. A number of defense genes are involved in the ROS and NO signaling pathway (Jones & Dangl, 2006). Recently, it has been shown that ornithine delta-aminotransferase (δO A T) and proline dehydrogenases (ProDH) are involved in ROS production in mitochondria and regulate defense responses such as HR cell death against nonhost pathogens (Senthil-Kumar & Mysore, 2012). The peroxisomal enzyme glycolate oxidase (GOX) has also been reported as an essential component of nonhost resistance. In Arabidopsis, GOX functions independently from NADPH oxidase (Respiratory burst oxidase homolog) and is involved in H2O2 accumulation and callose deposition in response to nonhost pathogen inoculation (Rojas et al., 2012). It is not clear whether COR can disrupt nonhost resistance responses mediated by δO A T, ProDH, and GOX.

Coronatine represses the expression of SA-related defense genes during nonhost resistance responses in tomato

In previous studies, it has been demonstrated that COR activates the expression of JA-inducible genes and suppresses SA-mediated defenses in tomato (Ishiga et al., 2009; Uppalapati et al., 2005; Uppalapati et al., 2007). To determine the expression profiles of SA- and JA-related genes upon inoculation with host or nonhost pathogens supplemented with COR, we inoculated tomato leaves with host pathogen P. syringae pv. tomato T1 and nonhost pathogen P. syringae pv. tabaci with or without COR. Expression profiles of SA-mediated signaling pathway genes, such as PR1a (pathogenesis-related protein 1a), PR1b (pathogenesis-related protein 1b), PR2b (pathogenesis-related protein 2b), and ICS1 (Isochorismate synthase 1), and JA-mediated signaling pathway genes, such as LoxD (Lipoxygenase D) and OPR3 (12-Oxophytodienoic acid reductase 3) were evaluated by real-time quantitative RT-PCR analysis at two time periods, 6 and 12 h post inoculation (hpi). Samples were not collected beyond 12 hpi, due to occurrence of cell death. As shown in Fig. 3, the expression of SA-mediated signaling pathway genes, including PR1a, PR1b, PR2 and ICS1, was more strongly induced by nonhost pathogen P. syringae pv. tabaci when compared to induction by host pathogen P. syringae pv. tomato T1. We further tested if the addition of exogenous COR resulted in a suppression of SA-mediated signaling pathway genes in P. syringae pv. tabaci-inoculated leaves. In COR-supplemented leaves, the expression of PR1a, PR1b, PR2 and ICS1 was significantly lower than nonhost pathogen alone inoculated leaves. In contrast, the expression of the JA-inducible genes, such as LoxD and OPR3, was more strongly induced in Psta-inoculated leaves supplied with COR when compared to Psta or COR alone. These results strongly support the hypothesis that COR functions to suppress the SA-mediated signaling pathway induction during nonhost disease resistance. In addition, COR may exploit the endogenous antagonistic interactions between JA and SA hormone signaling pathways, leading to inhibition of early plant defense mechanisms of nonhost disease resistance.

Figure 3 Expression profiles of SA- and JA-related defense genes in tomato leaves inoculated with nonhost pathogen P. syringae pv. tabaci (Psta) and host pathogen P. syringae pv. tomato T1 (Pst T1), and COR.

Tomato leaves were treated with distilled water (mock control), COR (100 ng/ ml) or inoculated with Psta, Psta with COR or Pst T1. The expression of genes encoding PR1a, PR1b, PR2b, ICS1, LoxD and OPR3 was evaluated by real-time quantitative RT-PCR. Bars represent the means ± standard deviation (SD) from three biological replications and three technical replications per experiment. Different letters and symbols above bars indicate significantly different values between treatments (P < 0.05, Student’s t test).

Conclusion

A number of studies have been reported regarding the role of COR in suppression of early plant defense responses like stomatal closure and induction of SA. However, it was not clear if COR can also suppress defense responses induced by nonhost pathogens. This study clearly demonstrates that COR suppresses nonhost HR, stomatal closure induced by nonhost pathogens and disrupts the hormonal defense signaling pathway activated during nonhost resistance. However, COR is not sufficient by itself to completely suppress nonhost resistance. This indicates that other downstream defense components are not targeted by COR during nonhost resistance. It warrants further investigation to understand the molecular mechanism of COR for suppression of early plant defense responses during nonhost resistance.

Supplemental Information

Supplemental Figure S1 Supplemental Figure S1

Development of disease symptoms by host pathogen P. syringae pv. tomato DC3000 in tomato. (A) Tomato leaves were spray-inoculated with P. syringae pv. tomato DC3000 (5 × 103 CFU/ml) and P. syringae pv. tabaci (5 × 103 CFU/ml), and imaged five days after inoculation. (B) The nonhost HR cell death by P. syringae pv. tabaci in tomato. Tomato leaves were syringe-infiltrated with host pathogen P. syringae pv. tomato DC3000 (6 × 102 CFU/ml) or nonhost pathogen P. syringae pv. tabaci (6 × 102 CFU/ml). The nonhost HR cell death was observed 48 h after infiltration.

Click here for additional data file.

Supplemental Table 1 Supplemental Table 1

Click here for additional data file.

Additional Information and Declarations

Competing Interests

Author Contributions

Kirankumar S. Mysore is an Academic Editor for PeerJ. The other authors do not have any competing interests.

Seonghee Lee conceived and designed the experiments, performed the experiments, analyzed the data, wrote the paper.

Yasuhiro Ishiga and Kristen Clermont performed the experiments.

Kirankumar S. Mysore conceived and designed the experiments, analyzed the data, wrote the paper.

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
