# Peer review of "Coronatine inhibits stomatal closure and delays hypersensitive response cell death induced by nonhost bacterial pathogens"

_PeerJ, doi:10.7717/peerj.34_

## Round 0.1 · original submission · Major Revisions

A point by point response to each comment provided by the two reviewers has to be given by the authors in a satisfactory way, before this manuscript may become acceptable.

Reviewer 1 ·

Basic reporting

Please specify 'a, b and c/d' in Figure2B and Figure 3D, Figure 4 at the figure legends.

Experimental design

How many biological and technical replicates for each sample are used for the real-time PCR? Ideally, three independent biological replicates and two technical replicates for each treatment are required to get meaningful statistical result. Please explain why 6h and 12h after inoculation were chosen to study the expression level?

Validity of the findings

It would be nice if the authors can discuss a few sentences about other factors that affect the defense mechanisms of nonhost disease resistance.

Reviewer 2 ·

Basic reporting

Summary
Please add a concluding sentence at the end stating the relevance of this study.

Introduction
1) There are no references cited in this section. Please add citations throughout the introduction as needed.

2) Third paragraph:
a. second sentence sounds ambiguous and author may wish to replace the word “non-adapted”. Is it “non-host” better?
b. Third sentence: spell the word syringae

Experimental design

Material and Methods

1) First paragraph (Plant and pathogen materials): details on bacterial culturing are missing.
a. Reference or composition of King’s B medium?
b. Which antibiotic and concentration was used?
c. Last two sentences should be deleted. Different types of inoculum were used (suspended in water or MES). It is unlikely that 1 ml of bacterial culture is enough for spraying a plant as mentioned in results.

2) Second paragraph (stomata assay): 100ng/ml (300 uM) of COR was used for stomatal assays. This is about 200 times more than the amount (0.5ng/ml or 1.5 uM) used by Zhao et al. 2003 and Melotto et al. 2006 in their assays. A dose response experiment should be included to justify the use of such high concentration of a purified chemical.

3) Procedure for spray-inoculation is not described. Authors should consider adding a section describing pathogenesis assay (inoculation, vehicle for inoculation, mock control, bacterial counts in planta, etc.).

4) Inhibition of non-host HR cell death:
a. reference for the RNAi line should be added or an explanation how this line was obtained.
b. What was the concentration of COR used for DAB staining? Reference for the procedure is missing.

5) Real-time quantitative PCR:
a. the description under this section is for RT-qPCR. Please revise the sub-heading accordingly.
b. Please describe the how fold expression was calculated (add the equations, the software used, or a reference).
c. I could not find Table 1 with primer sequences in the manuscript.

Validity of the findings

Results and Discussion
It is widely accepted that non-host bacteria causes HR in tomato, while the host bacteria does not. Therefore, Figure 1 repeats the literature.

It is widely accepted that stomatal closure is a functional output of plant’s defense against bacteria (host or non-host). This has been shown in innumerous plant pathosystems including tomato (the pathosystem used in this study). It has been also shown that COR re-opens stomata that have been closed by the non-host-specific PAMPs LPS, flg22 of different biological sources. Therefore, Figure 2 is a repetition of the literature.

Figure 3C: This is the only figure panel in this manuscript that convincingly shows COR-meditated suppression of HR in non-host plant. Yet, discussion about this figure is underdeveloped.

Figure 3D: The authors state that “The level of bacterial growth of nonhost pathogen P. syringae pv. tabaci (with COR) was similar to the host pathogen P. syringae pv. tomato DC3000 one day after inoculation (dpi), but the bacterial population of P. syringae pv. tabaci was reduced at 3 dpi, suggesting that COR is not sufficient by itself to completely suppress the defense mechanism(s) of nonhost disease resistance”. However, they do not consider the alternative hypothesis that Ps tabaci +COR population titers in the plant is similar to that of Pst DC3000 at day 1 because COR facilitates that penetration of tabaci by keeping most of stomata opened (especially by the extremely high level of COR used). Additionally, it is possible that Ps tabaci population declines at day 3 because COR is not stable for that longer in the plant and its effect fades off over time. As a side note, it is not clear whether bacterial enumeration reported refers to apoplastic population or total population (epiphytic + endophytic).

Figure 4 is an unnecessary repetition of the literature. COR is a non-host-specific toxin that up-regulates JA pathways and suppresses SA pathway. If any plant stress induces SA signaling, obviously by adding COR to the system, SA will be down-regulated; that is the action of COR at the cellular level. I would like to bring the authors attention to their data on ICS1 expression; mock treatment induces this gene in about 5 fold at 12h and COR alone does not. This clearly indicates that COR suppression of SA has nothing to do with what induces SA. COR simply keeps SA levels down and it is not specific no any stress.

Additional comments

Authors should consider adding line numbers to facilitate the review.
In general the manuscript is well written; however, important information is missing. For instance, the method section does not have enough details so that one can interpret the results (e.g. were plants surface-sterilized prior to bacterial count?). Three out of the four figures are repetition of published studies and the alternative explanations of the results are not presented (see comments on Figures 3D and 4) ; therefore, conclusions are weakly supported.

---

## Round 0.2 · accepted · Accept

The reviewer comments have been properly addressed, the manuscript is acceptable.

Reviewer 1 ·

Basic reporting

The revised version has answered the questions from both reviewers properly and should be accepted.

Experimental design

No Comments

Validity of the findings

No Comments

Additional comments

No Comments